# Growing Carbon Nanotubes In Situ Surrounding Carbon Fiber Surface via Chemical Vapor Deposition to Reinforce Flexural Strength of Carbon Fiber Composites

**DOI:** 10.3390/polym15102309

**Published:** 2023-05-15

**Authors:** Guangming Yang, Fei Cheng, Shihao Zuo, Jinheng Zhang, Yang Xu, Yunsen Hu, Xiaozhi Hu

**Affiliations:** 1School of Materials and Chemistry, Southwest University of Science and Technology, Mianyang 621010, Chinajhzhang2000@163.com (J.Z.); yxu2001@126.com (Y.X.); 2Engineering Research Center of Biomass Materials, Ministry of Education, Southwest University of Science and Technology, Mianyang 621010, China; 3School of Mechanical Engineering, Jiangsu University of Science and Technology, Zhenjiang 212003, China; yshu1995@126.com; 4Department of Mechanical Engineering, University of Western Australia, Perth, WA 6009, Australia

**Keywords:** carbon fiber reinforced polymer (CFRP), chemical vapor deposition (CVD), resin pre-coating (RPC), vertically aligned carbon nanotubes (VACNTs), integrated fiber bridging

## Abstract

This study employed novel joint treatments to strengthen the carbon fiber reinforced polymer (CFRP) composites. Vertically aligned carbon nanotubes (VACNTs) were prepared in situ on the catalyst-treated CF surface via the chemical vapor deposition (CVD) method, intertwining into three-dimensional fiber-nets and fully surrounding CF to form an integrated structure. The resin pre-coating (RPC) technique was further used to guide diluted epoxy resin (without hardener) to flow into nanoscale and submicron spaces to eliminate void defects at the root of VACNTs. Three-point bending testing results showed the “growing CNTs and RPC”-treated CFRP composites yielded the best flexural strength, a 27.1% improvement over the specimens without treatment, while the failure modes indicated that the original delamination failure was changed into “flexural failure” with through-the-thickness crack propagation. In brief, growing VACNTs and RPC on the CF surface enabled toughening of the epoxy adhesive layer, reducing potential void defects and constructing the integrated quasi-Z-directional fiber bridging at the CF/epoxy interface for stronger CFRP composites. Therefore, the joint treatments of growing VACNTs in situ via the CVD method and RPC technique are very effective and have great potential in manufacturing high-strength CFRP composites for aerospace applications.

## 1. Introduction

Carbon fiber reinforced polymer (CFRP), a novel composite with high strength/modulus to weight, fatigue resistance, excellent corrosion resistance, and designability [1,2,3], has been widely applied to important military and civilian fields [4,5,6,7,8,9,10] of aerospace, wind turbine, high-end cars, construction, etc. However, thin-walled structures of CFRP composite manufactured by 2D fiber fabrics are relatively weaker in the epoxy adhesive layer and in the thickness direction in contrast to those homogeneous materials that are isotropic or identical in all directions, and these typically result in interfacial cracking and delamination, thus causing premature failures under external complex loads [11,12].

Through-the-thickness (TTT) or Z-directional reinforcing methods (e.g., Z-pinning and stitching [13,14]) are the most effective to prevent delamination cracking at the ply interfaces, but they are also prone to causing deteriorations in the in-plane properties [15]. Evidently, mechanical Z-directional or TTT reinforcing methods will change fiber alignments of continuous carbon fibers and lead to local resin-rich regions. In addition, sparsely distributed short or micro-length flexible aramid fibers [16,17,18], chopped biological fibers [15,19,20], reticulated polyethersulfone fibers [21], various grades CNTs and carbon fibers [22,23,24], etc. can be premixed with epoxy to manufacture prepreg (fiber interleaving) or be directly and evenly dispersed into the epoxy adhesive for quasi-Z-directional toughening.

Those interleaving techniques using various fiber veils and films have been proven to improve compressive and flexural strength and fracture toughness. However, they do not seem to be successful in constructing Z-directional or quasi-Z-directional fiber bridging at the ply interface. Nanoscale-reinforced fiber itself is very difficult to gather and form approximately continuous fiber groups; it mainly works on the epoxy matrix toughening and lacks interlay structure improvement [25]. Large-size short aramid fiber can be extruded into resin-rich regions at ply interfaces, but it is almost unlikely to be embedded into the CF layer to build the bridging behavior [18]. Even if hierarchical aramid pulp fibers possess micro-fiber trunks and many nano-fiber branches, only a few free fiber ends are confirmed to be able to infiltrate and migrate the into shallow interface of the CF layer [26]. Direct external introduction of reinforcing fibers or additives has been developed for many years; it is very difficult to construct the physical connection and fiber bridging with the CF layer. Accurately preparing the integrated structure of reinforcing fiber and carbon matrix may be a theoretically valid solution and should be focused on because it is very likely to create and develop an innovative improvement process for manufacturing high-strength CFRP composites in industrial applications.

CNTs can grow on various substrates via chemical vapor infiltration (CVI), chemical vapor deposition (CVD), arc discharge, or laser ablation [27,28,29,30,31,32]. Among them, CVI and CVD are two effective methods to design and construct the specified structure of CNTs in situ on the matrix material. CNTs via CVD methods can only be grown on the matrix surface with no chemical reaction occurring to the matrix and no modifications to the related properties. However, growing CNTs via CVI can infiltrate into the superficial zone or interior of the matrix material, and it may react with the composition of the matrix, causing varying properties. In order to maintain the excellent mechanical properties of the CF, the CVD may be a more reliable method to manufacture high-strength CFRP composites, which was frequently studied in synthesizing CNTs. Ibrahim [33] adopted CVD to fabricate type-selected horizontally aligned single-walled CNTs. Zhang [34] proposed that temperature, catalyst kinds and sizes, carbon-metal adhesion, carbon concentration, and distribution are major factors for the nucleation and growth of CNTs. Those growing parameters of CNTs, including fiber diameter, fiber length, and areal density, can be controllably designed [35] to match diverse substrate surface conditions. It may be realized that vertically aligned CNTs with small diameter and long and entangled distributions can be grown on the CF surface via the CVD method.

Adopting the CVD method to synthesize CNTs on the CF surface has three advantages over interleaving techniques: (1) exhibiting good growth controllability; (2) easily constructing quasi-Z directional fiber bridging based on vertically growing CNTs; and (3) creating a physical connection with the CF to form a strong mechanical interlocking behavior. Thus, growing vertically aligned CNTs surrounding the CF surface is a novel and effective method to improve the interfacial adhesive bonding issue of interleaving techniques.

To our knowledge, there are very few studies conducted on growing CNTs on the CF surface via the CVD to improve the bonding strength of laminated CFRP composites, where no further surface treatment is used on the CNT-grown CF to improve the adhesive bonding surface condition of CF fabrics. In this study, the CVD method was employed to grow vertically distributed carbon nanotubes (VACNTs) in situ on a catalyst-treated CF surface. A simple resin pre-coating (RPC) technique was for the first time innovatively applied onto CF fabrics with VACNTs to eliminate potential void defects at the root of clustered CNTs; it was successful in preparing a stronger epoxy adhesive bonding interface to reinforce CFRPs. The flexural strength of CFRP composites before and after growing CNTs was examined to determine the reinforcement effect, and the internal cracks of damaged specimens after flexural tests were carried out to analyze the improving mechanism. Moreover, the micro-morphologies, pore distribution, and width of the CF with CNTs and interlayer appearances of CFRP composites were also characterized.

## 2. Composites Design, Preparation and Characterization

### 2.1. Optimized Design Concept of Interlayer Structure

Resin-rich regions and air bubbles are mainly interlayer defects of laminated CFRP composites, and weak bonding interface of CF/epoxy and lack of Z-directional or quasi-Z-directional reinforcing are dominant concerns: the combination of these factors is very likely to deteriorate the in-plane properties of CFRP composites. As demonstrated in Figure 1a,b, obvious resin-rich regions (several dozens of microns in thickness) can be observed between CF layers in the plain CFRP composite, which is prone to causing cracking and further propagating along the bonding interface of epoxy/CF without fiber bridging in the interfacial transition region as displayed in Figure 1d. Thus, constructing quasi-Z directional fiber bridging at the epoxy/CF interface is the key, and it can be achieved by growing VACNTs in situ on the CF surface. The evenly distributed CNTs with enough length are embedded into and will toughen the epoxy adhesive layer. It should be considered that CNT fiber may intertwine into small-size spaces that cause potential void defects, which could be eliminated using a simple RPC technique to improve the interfacial transition region. Therefore, the joint treatments of growing VACNTs and RPC can work theoretically and are desirable.

### 2.2. Starting Materials and CFRP Composites Preparing and Manufacturing

The CNT-reinforced CFRP composites were prepared as follows: (i) growing CNTs in situ on the carbon fiber surface; (ii) RPC treatment of CF fabric with CNTs; and (iii) compression molding of CFRP composites. The main starting materials were 3K plain woven carbon fiber fabrics (density 300 g/m^2^, bought from Shanghai Longchi Construction Technology Co., Ltd., Shanghai, China), 105 epoxy resin, and 206 slow hardeners (boiling point > 204 °C, purchased from West System Inc., Bay City, MI, USA), acetone (AR 99.5%, obtained from Chengdu Chiron Chemicals Co., Ltd. Chengdu, China), purple Fe(NO_3_)_3_·9H_2_O powder and red-brown Co(NO_3_)_2_·6H_2_O powder (AR 99%, both supplied by Shanghai Aladdin Biochemical Technology Co., Ltd., Shanghai, China).

A special three-dimensional fiber network structure on the CF surface was constructed by vertically growing CNTs via CVD as demonstrated in Figure 2. CF fabrics were immersed into mixed solutions of 1 mol/L Co(NO_3_)_2_ and 1 mol/L Fe(NO_3_)_3_ for 20 min to coat a layer of catalytic agent on the CF surface. Excessive water of catalytic-coated CF fabrics was fully evaporated by placing them into a dry oven at 200 °C for 1 h. The dried CF fabrics were laid into the tube furnace at 660 °C with an atmosphere of 100 mL/min N_2_ and 40 mL/min C_2_H_2_, conducting for 30 min to grow quasi-vertically-aligned CNTs.

RPC technique was applied to the CNT-growing CF fabrics to eliminate the void defects in the three-dimensional fiber network structure. RPC solution composed of 90 wt.% acetone and 10 wt.% epoxy resin (without hardener) was sprayed sufficiently onto CNT-growing CF fabrics, and resin-coated CF fabrics were formed after complete volatilization of acetone.

Final CFRP composites were processed by compression molding. CF fabrics with different surface conditions were adhered to by normal epoxy resin + hardener adhesive to manufacture primary CFRP composites with 10 plies CF and 9 plies epoxy adhesive interlays. They were molded at 10 MPa for 15 h and then placed into a dry oven at 60 °C for 96 h to obtain totally cured CFRP composites.

Completely cured CFRP composites were further manufactured for three-point bending tests. Besides the thickness of specimen itself, the dimension of the CFRP composite was designed and cut mechanically into approximately 90 mm × 13 mm (length × width) based on ASTM Standard D7264. All the important specimen parameters, including treatment methods of CF, panel thickness, and increase in thickness per epoxy layer, are listed in Table 1. The very small increases were obtained after the treatments of both growing VACNTs and growing VACNTs + RPC, showing that the ultrathin CNT interleaving was formed at the interlayer.

### 2.3. Composites Tests and Characterizations of Laminated CFRP Composites

The micro-morphologies of carbon fiber before and after growing VACNTs were observed using a scanning electron microscope instrument (SEM, ZEISS Gemini 300) at the voltage of 3.0 kV.

Three-point bending (3-P-B) tests of CFRPs were conducted using MTS CMT4104 universal testing machine equipped with a 10 kN load cell (Jinan MTS Test technology Co., Ltd., Jinan, China). The two-point span of the support beam was selected according to ASTM Standard D7264. The displacement mode was set to the constant rate of 2 mm/min, and the tests were instantly stopped once the load showed a precipitous decline.

The interlayer appearances of CFRP composites were revealed by the optical microscope (OM, WUMO WMJ-959, Shanghai WUMO Optical Instrument Ltd., Shanghai, China), and pictures were taken and processed with the matched WUMO WM-3000C.

The Brunauer–Emmett–Teller (BET) surface area was characterized by an automatic physical adsorption instrument (Quantachrome Instruments Autosorb-IQ US) with nitrogen adsorption and desorption. The test process lasted for 7 h and the desorption temperature of nitrogen was at 100 ℃.

Damaged specimens after 3-P-B tests were scanned using an X-ray microcomputed tomography (X-ray μCT) system (Versa 520, Zeiss, Pleasanton, CA, USA) for detecting the internal crack patterns. The Scout and Scan software (v11.1.5707.17179, Zeiss, Jena, Germany) was run at a constant voltage of 30 kV and a current of 67 μA to observe the post-peak crack and failure patterns.

## 3. Results and Discussions

### 3.1. Microstructure and Pore Analysis of Carbon Fiber after Growing VACNTs

Figure 3 shows the microstructures and corresponding microscopic model diagrams of untreated CF and the one after growing VACNTs in situ. Fiber trunks can be clearly observed in Figure 3a,b, and a few impurities resulting from transportation or sampling are attached to the CF surface. After the CVD treatment, several micron-long CNTs grow vertically surrounding the outside surface of the CF due to high graphitization; the CF trunk has been perfectly wrapped, so the composite structure seems to be greater in diameter. The encircled and intertwined CNTs are enough to construct a three-dimensional fiber net-structure and even create nanoscale and submicron spaces among CNT fibers on the CF surface [36]. Figure 4a presents the adsorption/desorption curves of CF with growing VACNTs, and the typical Ⅲ isotherm of the CF with growing VACNTs can be observed in Figure 4a. The pore width of the specimen located at the peak is 13.992 nm, exhibiting uniformly distributed mesopores [36] as displayed in Figure 4b. This indicates that nanoscale and submicron spaces were created on the CF surface by growing VACNTs in situ via the CVD, and they contributed to impregnating diluted resin solution and even constructing fiber bridging at the CF/epoxy bonding interface.

### 3.2. Flexural Strength of CFRP Composites with and without Growing VACNTs

Typical bending load and displacement curves of CFRP composites with three conditions are shown in Figure 5a; only the reflective one (the closest to the group average behavior) of each group is selected to be shown since a total of six specimens from each group were measured by 3-P-B tests. Greater failure displacements can be obtained with a further increase of treatments, exhibiting the reinforced CFRP composites after treatment. Their average flexural strength values were calculated by Equation (1) according to ASTM Standard D7264,
(1)$$\sigma =\frac{3PL}{2bh^{2}}$$
where, σ is the stress, *P* means the applied force, *L* signifies the span, *h* shows the beam thickness, and *b* shows the beam width. Apparently, the CFRP composites with VACNTs have higher flexural strength than the base plain, and the ones further coated by resin (without hardener) before mixing with normal epoxy resin + hardener adhesive yield the best strength of 718.86 MPa enhanced by 27.1% compared with the plain base strength, exhibiting a higher flexural strength or greater improvement than those in the literature [26,37]. This indicates that the joint treatments of growing VACNTs in situ via the CVD and RPC are very effective in laminated CFRP composite reinforcement.

Recently, ultrathin films and veils have been proven effective in improving methods for laminar CFRP composites. The greater interlayer thickness (over 30 μm) may contribute to reinforcing fracture toughness, but it is very likely to deteriorate the mechanical strength [38,39] as shown in Figure 5c, because thicker adhesive layers are easier to lead to the generation and propagation of microcracks. In comparison, the ultrathin interleaving beyond 30 μm, such as short chopped aramid fiber [11,16,17], and hierarchical aramid pulp [12,25,26], has been studied for flexural strength, compressive strength, and even elastic modulus enhancement of CFRP composites. In this study, the increased interleaving thickness due to growing VACNTs and RPC treatment is just an average of 3.9 μm (far below 30 μm); it creates the coequal or even higher enhancement effect, testifying once again it is the ultrathin films and veils rather than the thicker interleaving layer that enables strengthening of the laminar CFRP composites.

### 3.3. X-ray μCT Analysis of CFRP Composites after 3-P-B Testing

In order to reveal the internal structures and damage patterns of CFRP composites without mechanical sectioning, the specimens after the 3-P-B testing were scanned by X-ray μCT, and the results are presented in Figure 6. It can be clearly seen that the damage and cracks were initiated at the top side with compressive stresses for all the specimens under three different conditions. Cracks generated and mainly propagated along the CF interlayer to cause the delamination failure in the untreated specimen as depicted in Figure 6a. After growing VACNTs, the specimen also showed a few cracks propagating along the layer plane direction, but more cracks showed through the thickness to damage a multilayer CF as in Figure 6b. The specimen with growing VACNTs and RPC exhibited a through-the-thickness crack propagation, with almost no crack propagation occurring to the epoxy adhesive layer as observed in Figure 6c. It might be attributed to the fact that the fiber bridging was built by growing VACNTs and further RPC treatment to increase the delamination resistance, effectively preventing premature delamination. Higher compressive stresses needed to be applied to the specimen to cause the structure failure, so the flexural strength was improved. It should be noted that the cracks did not propagate to reach the tensile surface, namely, the bottom side for all three group specimens, illustrating that the bottom continuous carbon fibers were virtually undamaged, which explained that specimens could bounce back as soon as the applied load was removed.

Overall, the failure modes of bending-tested CFRP composites may be classified as compressive, tensile, and shear failure related to Mode-II delamination [22], which is decided theoretically by the maximum normal and shear stresses [40,41] in the 3-P-B testing based on the classical beam theory. The term “flexural failure” used in this study may be related to the ASTM Standard D7264.

### 3.4. Reinforcement Mechanism Analysis of VACNTs on Laminated CFRP Composites

Resin-rich regions between CF layers and weak epoxy/CF interfacial bonding [11] are the critical reasons leading to cracking under a lower external load; the cracks inevitably propagate along weak resin-rich regions and bonding interface of epoxy/CF, or even some potential air bubbles to cause the complete delamination failure of laminated CFRP composites, indicating a low flexural strength of untreated CFRP composites. The obvious reinforcement after the CVD treatment may attribute to a double role played by VACNTs in toughening epoxy resin in the adhesive layer and constructing fiber bridging at CF/epoxy bonding interface. However, better strength caused by the RPC treatment is due to the high viscosity of normal epoxy adhesives: they are difficult to flow into those nanoscale and submicron spaces as displayed in Figure 7(b2). However, the acetone-diluted resin solution (without hardener) with high liquidity can flow into these spaces and even reach the root of VACNTs or the CF surface, where residual composition can remain after the entire volatilization of acetone as demonstrated in Figure 7(c2). More importantly, the residue is still epoxy resin and its properties remain basically unchanged [12], which enables it to homogeneously mix with normal epoxy + hardener adhesive via diffusion to form a stronger bonding interface with minimal void defects as shown in Figure 7(c4) and also construct the quasi-Z directional fiber bridging. The improved interlayer adhesive bonding can resist cracking and inhibit crack propagation to reinforce the flexural strength of CFRP composites. In all, growing VACNTs via the CVD method is an effective technique to construct the physical connection between CF and reinforcing fiber, but further RPC treatment is still required to be conducted for processing a stronger bonding interface with fewer defects.

## 4. Conclusions

The novel joint treatments of “growing CNTs and RPC” have been verified to effectively reinforce the CFRP composites in this study. The CVD method was employed to grow CNTs in situ on the outer surface of catalyst-treated CF. SEM images showed massive vertically aligned CNTs intertwined into three-dimensional fiber-nets and fully surrounding CF to form the integrated structure. RPC technique was further applied to guide epoxy resin into nanoscale and submicron spaces created by VACNTs for eliminating void defects. Three-point bending testing results indicated the specimens with “growing CNTs and RPC” yielded the best flexural strength improved by 27.1% compared with the untreated specimens, and the failure modes showed a through-the-thickness crack propagation after the joint treatments instead of the original crack propagation along the epoxy adhesive layer to cause a delamination failure. The combination of “growing CNTs and RPC” could toughen the epoxy adhesive layer, construct the integrated quasi-Z-directional fiber bridging on the CF/epoxy interface, and avoid the void defects at the root of VACNTs for a stronger interlay bonding performance. Thus, growing VACNTs in situ via the CVD and RPC can be utilized together as an alternative applied to the manufacturing of high-strength CFRP composites. More importantly, the novel joint treatments may be also applied to other laminated fiber-reinforced (e.g., basalt fibers) plastic composites, or even fiber metal laminated composites to improve the interlayer structure and bonding strength.

## Figures and Tables

**Figure 1 polymers-15-02309-f001:**
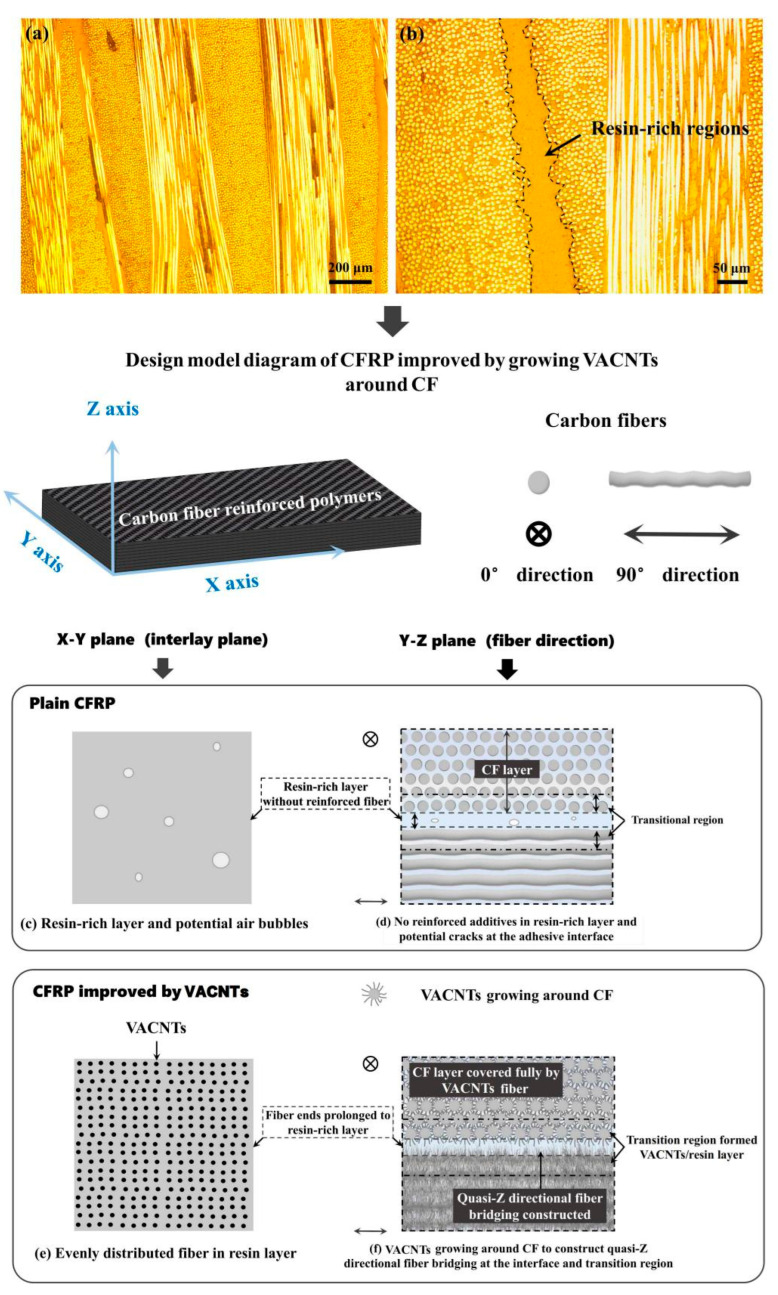
(**a**,**b**) Optical microscope photographs of plain CFRP composites, exhibiting resin-rich regions at the CF interlayer; (**c**–**f**) design model diagram of CFRP composites improved by growing VACNTs around CF via CVD method.

**Figure 2 polymers-15-02309-f002:**
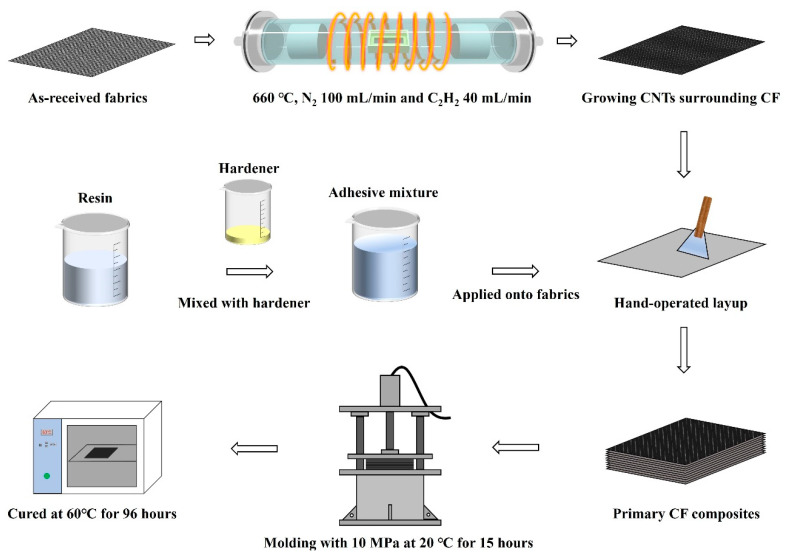
Preparation diagrams of reinforced CFRP composites with VACNTs growing in situ via the CVD method.

**Figure 3 polymers-15-02309-f003:**
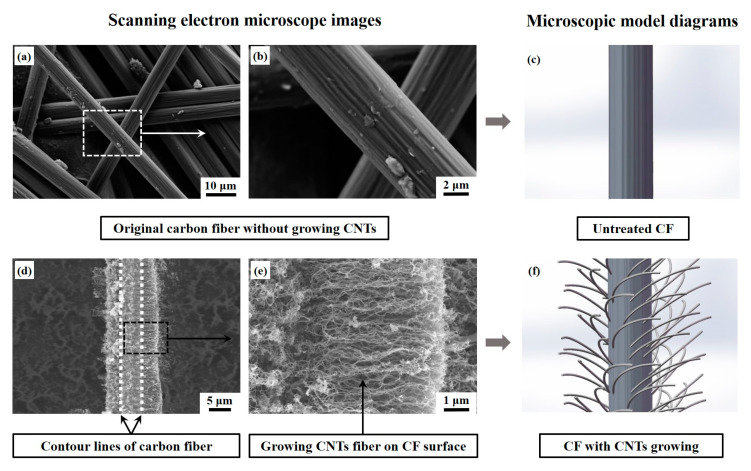
SEM images of CF: (**a**,**b**) untreated carbon fibers, (**d**,**e**) single carbon fiber after growing VACNTs via CVD, (**c**,**f**) corresponding microscopic model diagrams before and after growing VACNTs, respectively.

**Figure 4 polymers-15-02309-f004:**
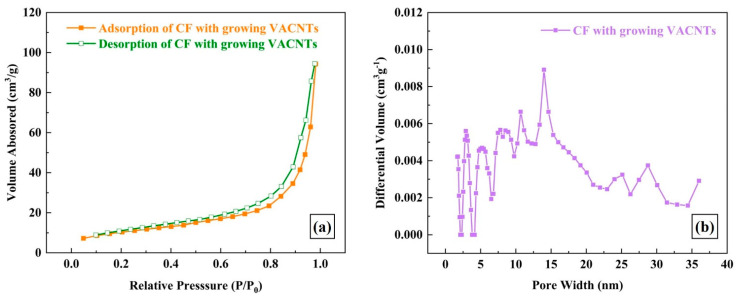
BET analysis results of CF with growing VACNTs in situ: (**a**) adsorption and desorption curves; (**b**) the pore width.

**Figure 5 polymers-15-02309-f005:**
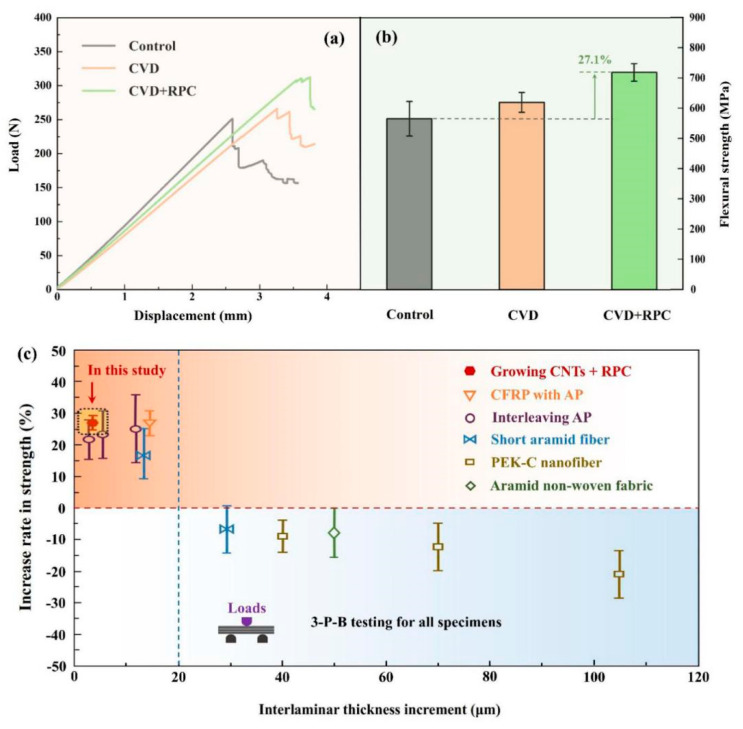
Three-point bend testing results of CFRP composites with different conditions: (**a**) reflective load-displacement curves and (**b**) average flexural strength (the standard deviations are displayed by error bars); (**c**) comparison of strength increase rates in this study with the results previously reported in the literature [12,17,26,38,39].

**Figure 6 polymers-15-02309-f006:**
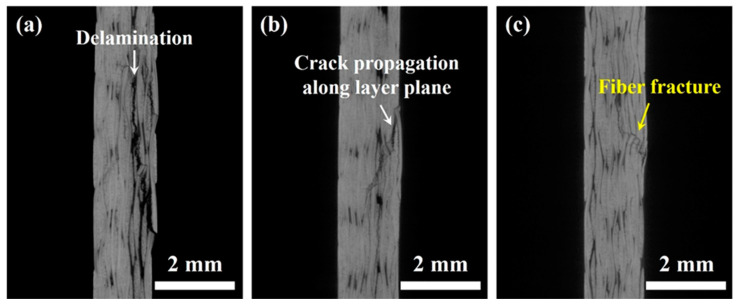
Internal microstructures of CFRP composites with three different conditions after failure (without mechanical polishing): (**a**) plain CFRP composites; (**b**) CFRP composites with growing VACNTs; (**c**) CFRP composites with growing VACNTs and RPC treatment.

**Figure 7 polymers-15-02309-f007:**
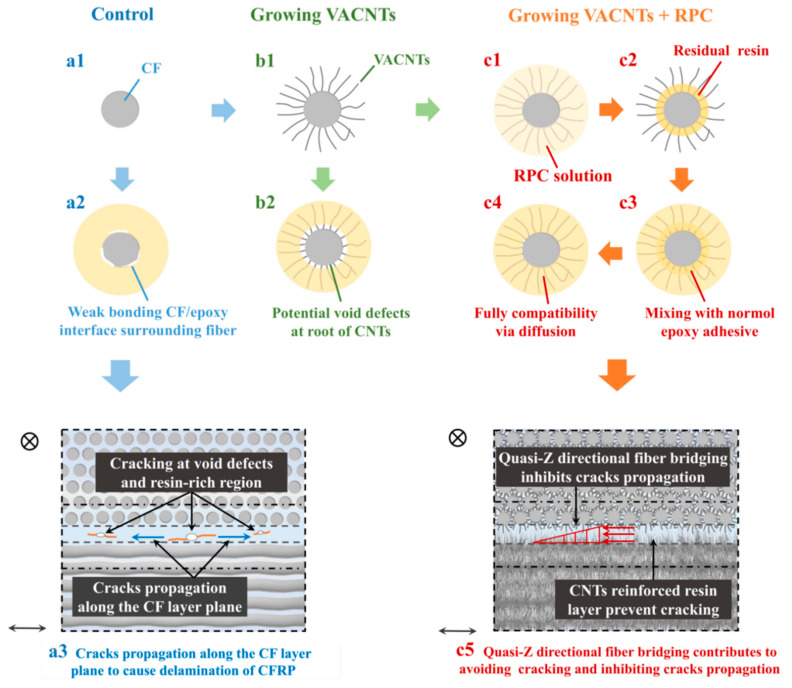
Reinforcement diagram of CFRP composite at the interlayer by comparing the untreated specimen (**a1**–**a3**), the specimen with growing VACNTs (**b1**,**b2**), and the specimen with growing VACNTs + RPC (**c1**–**c5**).

**Table 1 polymers-15-02309-t001:** Detailed parameters of CFRP composites after cutting mechanically into standard dimensions of three-point bending tests.

Specimens		Control	CVD	CVD + RPC
Treatments		Untreated	Growing CNTs in situ	Growing CNTs in situ + RPC before the normal adhesive
CF Ply number		10	10	10
Epoxy resin layer number		9	9	9
Specimen number of each group		6	6	6
Length (mm)		94.8	95.1	94.7
Width (mm)		13.1	12.9	13.1
Thickness (mm)	Average	1.603	1.628	1.638
	Standard derivation	0.178	0.181	0.182
Increase in thickness per epoxy layer (μm)		--	2.7	3.9

## Data Availability

Not applicable.

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
