# Peer review of "Growing Carbon Nanotubes In Situ Surrounding Carbon Fiber Surface via Chemical Vapor Deposition to Reinforce Flexural Strength of Carbon Fiber Composites"

_polymers, 2023, doi:10.3390/polym15102309_

Round 1

Reviewer 1 Report

My evaluation report is attached below. The final decision will be made after the revision.

Author Response

Reviewer #1: After a comprehensive review of the manuscript “Growing Carbon Nanotubes In-situ Surrounding Carbon Fiber Surface via Chemical Vapor Deposition to Reinforce Flexural Strength of Carbon Fiber Composites”, Topic and paper are interesting. There is no doubt in the originality of the work but I think the manuscript should be improved. I’ll try to clear my point through the following comments:

1. In the abstract, you must give results of failure modes, load-carrying capacities, energy dissipation capacities, peak amplitudes, fatigue strength, etc. % are needed. The introduction should be improved. More general literature review is needed and novelty of the research should be stated clearly. I think it should be added a new section which is “research significance”. In other words, the novelty and importance of the research should be clarified using up-to-date researches around the globe. The differences from previous studies must be specified clearly. Experiences from all over the world could help authors for performing a comprehensive research. You may also add flow chart about your study. The importance of using different materials for supporting should be included using followings: Experimental and Analytical Investigation of Flexural Behavior of Carbon Nanotube Reinforced Textile Based Composites; Experimental investigation of shear capacity and damage analysis of thinned end prefabricated concrete purlins strengthened by CFRP composite; Numerical investigation of the parameters influencing the behavior of dapped end prefabricated concrete purlins with and without CFRP strengthening; Experimental analysis of reinforced concrete shear deficient beams with circular web openings strengthened by CFRP composite; Shear strengthening of reinforced concrete T-beams with anchored and non-anchored CFRP fabrics; Behavior of CFRP-strengthened RC beams with circular web openings in shear zones: Numerical study; Optimum amount of CFRP for strengthening shear deficient reinforced concrete beams; The effects of eccentric web openings on the compressive performance of pultruded GFRP boxes wrapped with GFRP and CFRP sheets; Experimental analysis of shear deficient reinforced concrete beams strengthened by glass fiber strip composites and mechanical stitches; Compressive behavior of pultruded GFRP boxes with concentric openings strengthened by different composite wrappings; Effect of the GFRP wrapping on the shear and bending Behavior of RC beams with GFRP encasement; Strengthening of shear-critical reinforced concrete T-beams with anchored and non-anchored GFRP fabrics applications; Buckling Analysis of CNT Reinforced Polymer Composite Beam Using Experimental and Analytical Methods; Shear strengthening of reinforced concrete beams with minimum CFRP and GFRP strips using different wrapping technics without anchoring application; Numerical and analytical investigation of parameters influencing the behavior of shear beams strengthened by CFRP wrapping.

Thanks for your comments to improve the quality of manuscript. Indeed, the sections of Abstract and Introduction need to be revised to exhibit our study well, we have made some revisions in the resubmitted manuscript. In addition, we have also cited some publications as you recommended to support our study. Thanks again for your comments.

2. What is the limit for stopping the loading process? Why was the four-point loading not chosen? Please clarify.

Thanks for your comment. The 3PB tests of CFRP composites were instantly stopped once the load showed a precipitous decline as exhibited in “2.3. Composites characterization”. As is known to us, the 3PB tests are relatively simple and suitable for the laminated CFRP composites with a thin thickness. The concern about the vertical shear force may be solved by increasing the span-to-thickness ratios as recommended by ASTM D7264 (Standard Test Method for flexural Properties of Polymer Matrix Composite Materials). In past most of publications, 3PB tests were applied to laminated CFRP composites instead of four-point bending test. Therefore, 3PB tests were used in this study to characterize the flexural strength.

3. What is the criterion of authors for choosing the tested specimens? Using different specimens or even different scales could jeopardize the results of the research or not? I think it is better to choose these specimens from a constructed or at least a designed in building.

Thanks very much for your comment. This study conducted the novel joint treatments to reinforce the carbon fiber reinforced polymer  (CFRP) composites, the test parameters like dimensions, span of support beam tested specimens were selected according to ASTM D7264 (Standard Test Method for flexural Properties of Polymer Matrix Composite Materials) as mentioned in 2.3. Composites characterization. The tested specimens are CFRPs with only very tiny difference in the thickness, other parameters are totally based on ASTM D7264, therefore, the research results could not be jeopardized.    

4. The conclusion part is only a summary of the results. We need technical and general advises here which can be used by others (both researches and engineers).

Thanks for your valuable suggestion. We have revised the conclusion and added some advises for both researches and engineers in the resubmitted manuscript.

5. The work is a good report of experiments investigation, but which are the lessons learnt? The authors have to clarify before acceptance.

Thanks for your valuable comments. We have made the revisions to exhibit the research value and contribution to development of study and industry in the resubmitted manuscript.

Reviewer 2 Report

Dear Authors,

I studied your manuscript entitled "Growing Carbon Nanotubes In-situ Surrounding Carbon Fiber Surface via Chemical Vapor Deposition to Reinforce Flexural Strength of Carbon Fiber Composites". I recommend a major revision before further consideration for publication in the Polymers. If the revisions are perfectly made, the manuscript would be acceptable.

1) The major concern for this paper is the lack of novelty. With a quick search we can see that many similar studies have been done (https://doi.org/10.1016/j.msea.2010.11.067, https://doi.org/10.1016/j.carbon.2012.03.023, https://doi.org/10.1039/C6NR06479E, https://doi.org/10.1016/S1003-6326(14)63452-X, etc.). Your research findings should be compared with those of the recent literature review on "Growing Carbon Nanotubes on the Surface of Carbon Fibers" from different authors and years.

2) You should present and discuss some more analyses that evaluate the composite properties.

3) English language needs some polishing since some terms are vague.

English language needs some polishing since some terms are vague.

Author Response

Dear Authors,

I studied your manuscript entitled "Growing Carbon Nanotubes In-situ Surrounding Carbon Fiber Surface via Chemical Vapor Deposition to Reinforce Flexural Strength of Carbon Fiber Composites". I recommend a major revision before further consideration for publication in the Polymers. If the revisions are perfectly made, the manuscript would be acceptable.

1) The major concern for this paper is the lack of novelty. With a quick search we can see that many similar studies have been done (https://doi.org/10.1016/j.msea.2010.11.067, https://doi.org/10.1016/j.carbon.2012.03.023, https://doi.org/10.1039/C6NR06479E, https://doi.org/10.1016/S1003-6326(14)63452-X, etc.). Your research findings should be compared with those of the recent literature review on "Growing Carbon Nanotubes on the Surface of Carbon Fibers" from different authors and years.

Thanks for the valuable and helpful comment of reviewer to improve the quality of manuscript. We have made a comparison between the used method with those literatures on "Growing Carbon Nanotubes on the Surface of Carbon Fibers" as you mentioned, all the revisions have been exhibited in the resubmitted manuscript.

2) You should present and discuss some more analyses that evaluate the composite properties.

Thanks very much for your useful comment. Indeed, some important information in “Result and discussion” section was likely to be relatively lacked, we have added more discussions to evaluate the composite properties as you suggested in the revised manuscript.

3) English language needs some polishing since some terms are vague.

Thanks very much. We have reviewed the whole manuscript carefully, and tied our best to corrected the mistakes and vague terms as you mentioned. All the revisions have been shown in the resubmitted manuscript.

Reviewer 3 Report

The manuscript discusses the use of the chemical vapor deposition method to grow vertically aligned carbon nanotubes (VACNTs) on catalyst-treated carbon fiber (CF) surfaces, applying resin pre-coating techniques onto CF fabrics with VACNTs to reduce void defects at the root of clustered carbon nanotubes. The manuscript’s outline is well organized. The title and abstract are appropriate for the content of the text. The introduction section can be expanded and improved. The following are specific comments

1. What is the effect of VACNTs on the strength or reinforcement of CF? What is the role of VACNTs in toughening epoxy resin in the adhesive layer and constructing fiber bridging at the CF/epoxy bonding interface?

2. I don’t consider the topic original but relevant in the field. However, the research work shows significant contributions to the field.

3. This work proved the in-situ growth of VACNTs using CVD and RPC, which may be synergistically applied to develop strong/tough CFRPs.

4. In my opinion, the methodology of the study is satisfactory.

5. The conclusion is consistent with the results.

6. The references are appropriate.

7. Figure 4 lacks error bars. If the measurement is replicated, the authors need to include the error bars.

Author Response

Reviewer #3

The manuscript discusses the use of the chemical vapor deposition method to grow vertically aligned carbon nanotubes (VACNTs) on catalyst-treated carbon fiber (CF) surfaces, applying resin pre-coating techniques onto CF fabrics with VACNTs to reduce void defects at the root of clustered carbon nanotubes. The manuscript’s outline is well organized. The title and abstract are appropriate for the content of the text. The introduction section can be expanded and improved. The following are specific comments.

1. What is the effect of VACNTs on the strength or reinforcement of CF? What is the role of VACNTs in toughening epoxy resin in the adhesive layer and constructing fiber bridging at the CF/epoxy bonding interface?

Thanks for your constructive comments to help improve the level of our paper. Actually, thin-walled structures of CFRP composite manufactured by 2D fiber fabrics are relatively weaker in epoxy adhesive layer and in the thickness direction in contrast to those homo-geneous materials performing isotropic or identical in all directions, typically resulting in interfacial cracking and delamination to cause premature failures under external complex loads. In this study, VACNTs were grown in-situ on the CF surface via CVD methods, which contributed to creating a condition on constructing fiber bridging behaviors at the interlayer of laminated CFRP composites. The direct effect on the strength or reinforcement of CF was not exhibited, growing VACNTs can have important influences on toughening epoxy resin in adhesive layer and constructing fiber bridging at CF/epoxy bonding interface for a stronger bonding strength. For toughening epoxy resin, the growing VACNTs around 10 μm in length observed in SEM images can embed into epoxy resin in the adhesive layer and perform the toughening effect on brittle epoxy resin to avoid cracking under a low external force. For constructing fiber bridging at the CF/epoxy bonding interface, manufactured CFRP composites had the interfacial transition region between CF and epoxy adhesive, complete adhesion or defects were very likely to be formed at the interface or shallow CF layer, causing cracks might generate and propagate along the bonding interface for delamination. The growing VACNTs enabled to construct fiber bridging at the CF/epoxy bonding interface with the joint treatments of RPC, the fiber bridging could inhibit the crack propagation, and it needed the higher load or energy absorption to damage the structure, therefore, the flexural strength of CFRPs could be reinforced.

2. I don’t consider the topic original but relevant in the field. However, the research work shows significant contributions to the field.

Thanks for your comments.

3. This work proved the in-situ growth of VACNTs using CVD and RPC, which may be synergistically applied to develop strong/tough CFRPs.

Thanks a lot.

4. In my opinion, the methodology of the study is satisfactory.

Thanks very much.

5. The conclusion is consistent with the results.

Thanks for your kind comments.

6. The references are appropriate.

Thanks for your positive comments.

7. Figure 4 lacks error bars. If the measurement is replicated, the authors need to include the error bars.

Thanks for your kind remind. Fig. 4 presents the adsorption/desorption curves and the pore width of CF with growing VACNTs. The test was conducted only one time, but many fibers growing CNTs were measured in a single nitrogen adsorption and desorption test to ensure that the obtained results were valid. Therefore, the error bars were not included.

Author Response

Reviewer #4

It’s a very interesting and valuable study on growing Carbon Nanotubes surrounding Carbon Fiber. Here are some comments to improve the manuscript:

(1) Fig 6 is showing a comparison of the internal microstructure for different samples after failure. It would be interesting to show the structure before and after the 3-point test as well.

Thanks for your suggestion. The internal microstructure of different specimens before 3-P-B tests have been scanned, and they did not show the substantial information to support the manuscript content since no any crack could be observed, therefore, they were not exhibited in Fig. 6 after careful consideration.

(2) The level of epoxy crosslinking is also an important factor to make the comparison. I suggest making another sample without any reinforcement to show the baseline of the mechanical properties, first. Also, developing another series of the test for different levels of epoxy crosslinking and reduce the comparison.

Thanks for your useful comment. It is indeed very necessary to show the baseline (without any reinforcement) of the mechanical properties. Actually in Fig. 5 (b), the specimen group of “control” is the specimens without any reinforcement (namely the base strength), total 6 specimens of each group were measured for calculating the flexural strength to ensure the repeatability of experimental results.

Round 2

Reviewer 1 Report

The manuscript may be accepted in its current form.

Minor editing of English language required

Reviewer 2 Report

Dear Authors,

Thank you for considering my comments. I have recommended the publication of your article as is.